# DOA Tracking Based on Unscented Transform Multi-Bernoulli Filter in Impulse Noise Environment

**DOI:** 10.3390/s19184031

**Published:** 2019-09-18

**Authors:** Sun-yong Wu, Jun Zhao, Xu-dong Dong, Qiu-tiao Xue, Ru-hua Cai

**Affiliations:** 1School of Mathematics and Computing Science, Guilin University of Electronic Technology, Guilin 541004, China; wsy121991@guet.edu.cn (S.-y.W.); jun325709@163.com (J.Z.); 18871347936@163.com (X.-d.D.); ruhuac@guet.edu.cn (R.-h.C.); 2Guangxi Information Science Experiment Center, Guilin 541004, China

**Keywords:** direction-of-arrival (DOA) tracking, impulse noise, Multi-Bernoulli filter, particle filtering

## Abstract

Aiming at the problem of multiple-source direction of arrival (DOA) tracking in impulse noise, this paper models the impulse noise by using the symmetric *α* stable (S*α*S) distribution, and proposes a DOA tracking algorithm based on the Unscented Transform Multi-target Multi-Bernoulli (UT-MeMBer) filter framework. In order to overcome the problem of particle decay in particle filtering, UT is adopted to select a group of sigma points with different weights to make them close to the posterior probability density of the state. Since the *α* stable distribution does not have finite covariance, the Fractional Lower Order Moment (FLOM) matrix of the received array data is employed to replace the covariance matrix to formulate a MUSIC spatial spectra in the MeMBer filter. Further exponential weighting is used to enhance the weight of particles at high likelihood area and obtain a better resampling. Compared with the PASTD algorithm and the MeMBer DOA filter algorithm, the simulation results show that the proposed algorithm can more effectively solve the issue that the DOA and number of target are time-varying. In addition, we present the Sequential Monte Carlo (SMC) implementation of the UT-MeMBer algorithm.

## 1. Introduction

Multi-target Direction of Arrival (DOA) estimation is an essential issue in array processing and has a wide range of applications in source location, radar, sonar, and wireless communications [1,2]. Sparse representation and compressive sensing methods are used for DOA estimation of coprime array [3,4,5,6], while these methods are only applied in the case where the sources are stationary. In addition, difficulties also arise from the uncertainties of the source dynamics: the source may be moving or static. Thus, it is significant to extend the static DOA estimation algorithm to the dynamic DOA tracking algorithm.

The representative dynamic DOA tracking algorithms include the subspace tracking algorithm and the particle filter (PF) algorithm. The subspace tracking algorithm includes Projection Approximation Subspace Tracking (PAST) [7] and the Projection Approximation Subspace Tracking with Deflation (PASTD) [8]. In essence, these algorithms transform the determination of the eigensubspace into solving an unconstrained optimization problem, and combine the recursive least squares (RLS) theory to achieve effective tracking of the eigensubspace of time-varying sources. However, the RLS method is very sensitive to impulse noise, and the PAST algorithm’s subspace tracking performance will degrade sharply in the impulse noise environment [9,10,11]. In an army of acoustic applications, such as underwater and room acoustic signal processing, the noise environment is non-Gaussian and is impulsive in nature [12,13]. Under investigation, it was found that *α* stable distribution (0 < *α* ≤ 2) is a suitable noise model to describe this type of noise [14]. In recent years, DOA estimation technology in impulse noise environment has developed rapidly [15,16,17]. The PF algorithm based on Bayesian recursive estimation can solve the target tracking problem by utilizing a priori DOA and current measurement information [18]. In [19], the author considers the particle filtering method to estimate the single target DOA by using the spatial spectral function based on FLOM matrix as the likelihood function in the impulse noise environment. However, those algorithms needs to know the number of targets in advance and cannot deal with the estimation problem of the time-varying sources DOA.

In practical applications, such as submarine tracking and sonar positioning, the number of the sources are dynamic. Mahler introduced the concept of random finite set (RFS) in [20]. A tutorial on Bernoulli filters is introduced in [21]. A track-before-detect (TBD) Bernoulli filter based on RFS is proposed for DOA tracking in single dynamic system in [22], but it cannot solve the DOA tracking in multiple target dynamic system. The Multi-target Multi-Bernoulli (MeMBer) filtering [23] is a filter developed under the RFS framework. The advantage is that it operates on the dimensions of a single target space, thus avoiding the computational complexity and data association problems of the joint filter. Choppala P B et al. studied the Bayesian multi-target tracking problem based on phased array sensor, and proposed the MUSIC spatial spectral as a pseudo-likelihood in the Multi-Bernoulli filter in [24]. However, the shortcoming of this algorithm is that impulse noise is not considered, and Gaussian noise model is not appropriate in practical applications.

Based on the above analysis, a particle filter algorithm of DOA tracking for Unscented Transform MeMBer (UT-MeMBer) in an impulse noise environment is proposed. UT is used to construct a new important density function, which makes the estimation accuracy higher when the particle degenerates. Since particles close to the real state are more likely to output a larger spatial spectral response, the magnitude of the spatial spectral response is used as a feature of pseudo-likelihood. Based on the FLOM matrix, this paper uses FLOM matrix to substitute the covariance matrix to obtain the corresponding MUSIC spatial spectrum as the particle likelihood function. Further exponential weighting can increase the weight of the particles, making resampling more efficient. The main advantage of the tracking algorithm is that the number and state of the target can be accurately tracked when the number and state of the sources are unknown in impulse noise environment.

The rest of the paper is organized as follows. In Section 2, the problem of the DOA tracking in impulsive noise environment is described. In Section 3, we outline the Multi-Bernoulli’s Bayesian theory of DOA tracking. An improved algorithm for likelihood functions is introduced in Section 4. The UT-MeMBer DOA particle filter tracking algorithm is given in Section 5. We then show our simulation results in Section 6 and conclusion in Section 7.

## 2. Problem Formulation

### 2.1. Array Signal Model

Consider the case of P narrow farfield signals $s_{p}\left(t\right),p=1,2,\cdots P$ with different DOA $\theta_{1},\theta_{2},\ldots ,\theta_{P}$ arriving at a uniform linear array (ULA) with *M* sensors at discrete time *t*. The DOA of the *p*th source can be written as $\theta_{p}$. The received signal of the arrays can be expressed as
(1)Z(t)=A(θ)S(t)+N(t)
where $N_{M\times 1}\left(t\right)={\left[n_{1}\left(t\right),n_{2}\left(t\right),\ldots ,n_{M}\left(t\right)\right]}^{T}$ represents the impulsive noise vector which is not correlated with signals. $Z_{M\times 1}\left(t\right)={\left[z_{1}\left(t\right),z_{2}\left(t\right),\ldots ,z_{M}\left(t\right)\right]}^{T}$ is the measurement at time *t*, $A_{M\times P}\left(t\right)={\left[a\left(\theta_{1}\right),a\left(\theta_{2}\right),\ldots ,a\left(\theta_{P}\right)\right]}^{T}$ is array manifold, $S_{P\times 1}\left(t\right)={\left[s_{1}\left(t\right),s_{2}\left(t\right),\ldots ,s_{P}\left(t\right)\right]}^{T}$ denotes the acoustic sources matrix, and
(2)$$a\left(\theta_{p}\right)=\left[1,e^{-j\frac{2\pi }{\lambda }d\sin\theta_{p}},\ldots ,e^{-j\frac{2\pi }{\lambda }\left(M-1\right)d\sin\theta_{p}}\right]$$
is the steering vector with λ denoting the wavelength of the carrier, and d is the array space. 

### 2.2. α Stable Distribution

Most of the traditional research methods estimating the DOA are based on Gaussian noise models. In practical situations, such as radar echo and low-frequency atmospheric noise, they consist of impulse noise with a short duration and large amplitude. The performance of the algorithm will drop significantly when the Gaussian noise model is still modeled in an impulse noise environment. The *α* stable distribution is a good example of such a type with significant spike noise and a Gaussian distribution. The *α* stable distribution’s probability function does not have the closed form, which can be conveniently described by its characteristic function as
(3)$$\phi \left(t\right)=e^{\{jat-\gamma {\left|t\right|}^{\alpha }\left[1+j\beta \mathrm{sgn}\left(t\right)\varpi \left(t,\alpha \right)\right]\text{\}}}$$
where
(4)$$\varpi =\left\{ \begin{array}{l} \tan\frac{\alpha \pi }{2},\alpha \neq 1\\ \frac{2}{\pi }\log\left|t\right|,\alpha =1 \end{array}\right.$$
(5)$$\mathrm{sgn}\left(t\right)=\left\{ \begin{array}{l} 1,t>0\\ 0,t=0\\ -1,t<0 \end{array}\right.$$

*α* is the characteristic exponent, whose size can affect the degree of impulse and the range is 0<α≤2. γ is a dispersion parameter whose mean is consistent with the variance of the Gaussian distribution. β is a symmetric parameter, and the distribution at β=0 is a symmetric *α* stable (S*α*S) distribution. a is the positional parameter. When α=2,β=0, it is a Gaussian distribution model. When α=1,β=0, it is the Cauchy distribution model. When α=1/2,β=−1, it is the Pearson distribution model. A crucial difference between the Gaussian distribution and the *α* stable distribution is that the latter does not have second-order statistics so that its covariance is inaccurate.

## 3. MeMBer Bayesian Theory of DOA Tracking

### 3.1. Multi-Target Bayesian Theory

Assume that the state of the sources at time *k* is $x_{k}={\left[\theta_{k},{\dot{\theta }}_{k}\right]}^{T}$, where $\theta_{k}$ is the DOA and moves at a speed of ${\dot{\theta }}_{k}$ rad/s. The state and number of sources are changing at time *k* + 1, which can be described by RFS. From [20], the sources state set in multiple sources tracking can be regarded as an RFS, namely
(6)$$X_{k}=\left\{x_{k,1},\cdots ,x_{k,P\left(k\right)}\right\}$$
where $X_{k}$ represents a set of sources at time *k*, and the element of the set may be one or more or null. $Z_{k}$ denotes the measurement set generated by all sources received time *k*, and the element is only one.

Single-target Bayesian filtering can be extended to multi-target tracking by modeling the above source states and measured values. The single target posterior probability density function (pdf) $p_{k|k}\left(x_{k}|Z_{1:k}\right)$ is replaced by the joint multi-target posterior $p_{k|k}\left(X_{k}|Z_{1:k}\right)$. The Bayes joint filter recursion includes two stages: prediction and update. The prediction and update at time *k* in [24] are
(7)$$p_{k|k-1}\left(X_{k}|Z_{1:k-1}\right)=\int f_{k|k-1}\left(X_{k}|X_{k-1}\right)p_{k-1|k-1}\left(X_{k-1}|Z_{1:k-1}\right)\delta X_{k-1}$$
and
(8)$$p_{k|k}\left(X_{k}|Z_{1:k}\right)=\frac{g\left(Z_{k}|X_{k}\right)p_{k|k-1}\left(X_{k}|Z_{1:k-1}\right)}{\int g\left(Z_{k}|X_{k}\right)p_{k|k-1}\left(X_{k}|Z_{1:k-1}\right)\delta X_{k}}$$
where δ is the set integral and $Z_{1:k-1}$ represents all the measurement sets up to time k−1. $g\left(Z_{k}|X_{k}\right)$ is a multi-target joint likelihood function and $f_{k|k-1}\left(X_{k}|X_{k-1}\right)$ is a multi-target state transition probability density function. $p_{k|k-1}\left(X_{k}|Z_{1:k-1}\right)$ represents the multi-target joint prediction probability density and $p_{k|k}\left(X_{k}|Z_{1:k}\right)$ is the multi-target joint posterior probability density function. 

### 3.2. Multi-Target Multi-Bernoulli Filter

A Bernoulli set X has a probability 1−r of being a null set, and has a probability r of containing a single element x that is distributed via a pdf s(·). The probability of a Bernoulli RFS can be expressed in [21] as
(9)$$\pi \left(X\right)=\{\begin{array}{c} 1-r,\;X=\emptyset \\ rs\left(X\right),\;X=\left\{x\right\}\\ 0,\;other \end{array}$$

A Multi-Bernoulli RFS X can be considered as union of a fixed number of independent Bernoulli sets that have existence probability $r^{\left(j\right)}\in \left(0,1\right),j=1,\ldots J$ and the pdf $s^{\left(j\right)}$, such that
(10)$$X=\overset{J}{\underset{j=1}{\cup }}X^{\left(j\right)}$$
where the *j* th Bernoulli set is described by its two parameters: the existence probability $r^{\left(j\right)}$ and the pdf $s\left(X^{\left(j\right)}\right)$. So a Multi-Bernoulli RFS can be characterized by a posterior parameter set ${\left\{\left(r_{k|k}^{\left(j\right)},s_{k|k}\left(X_{k}^{\left(j\right)}\right)\right)\right\}}_{j=1}^{J_{k}}$, where $J_{k|k}$ indicates the number of sources. $Z_{k}={\left[z_{1,k},z_{2,k},\ldots ,z_{M,k}\right]}^{T}$ denotes the sensor measurement data and $Z_{k}\in Z$, in which Z is the measurement space of the sensor. Target birth and survival are determined by birth probabilities $p_{b,k}\left(X_{k}\right)$ and survival probabilities $p_{s,k}\left(X_{k}\right)$, respectively. The source motion model is represented by the transition probability density $f_{k|k-1}\left(X_{k}|X_{k-1}\right)$, and the prior probability of Multi-Bernoulli is described as
(11)$$p\left(X_{k-1}|Z_{1:k-1}\right)\approx {\left\{r_{k-1|k-1}^{\left(j\right)},s_{k-1|k-1}\left(X_{k-1}^{\left(j\right)}\right)\right\}}_{j=1}^{J_{k-1}}$$

According to Equation (7), the prediction part can be described as
(12)$$\begin{array}{l} p\left(X_{k}|Z_{1:k-1}\right)\approx {\left\{{\hat{r}}_{k|k-1}^{\left(j\right)},{\hat{s}}_{k|k-1}\left(X_{k}^{\left(j\right)}\right)\right\}}_{j=1}^{J_{k|k-1}}\\ \;={\left\{r_{P,k|k-1}^{\left(j\right)},s_{P,k|k-1}\left(X_{k|k-1}^{\left(j\right)}\right)\right\}}_{j=1}^{J_{P,k|k-1}}\cup {\left\{r_{B,k}^{\left(j\right)},s_{B,k}\left(X_{k}^{\left(j\right)}\right)\right\}}_{j=1}^{J_{B,k}} \end{array}$$
where
(13)$$\begin{array}{l} {\hat{r}}_{k|k-1}^{\left(j\right)}=\left(1-r_{k-1|k-1}^{\left(j\right)}\right)\cdot \int p_{b,k}\left(X_{k}^{\left(j\right)}\right)s_{k-1|k-1}\left(X_{k-1}^{\left(j\right)}\right)dX_{k-1}^{\left(j\right)}\\ \;+r_{k-1|k-1}^{\left(j\right)}\cdot \int p_{s,k}\left(X_{k-1}^{\left(j\right)}\right)s_{k-1|k-1}\left(X_{k-1}^{\left(j\right)}\right)dX_{k-1}^{\left(j\right)} \end{array}$$
(14)$$\begin{array}{l} {\hat{s}}_{k|k-1}\left(X_{k|k-1}^{\left(j\right)}\right)=\frac{p_{s,k}\left(X_{k-1}^{\left(j\right)}\right)r_{k-1|k-1}^{\left(j\right)}\cdot \int f_{k|k-1}\left(X_{k}^{\left(j\right)}|X_{k-1}^{\left(j\right)}\right)S_{k-1|k-1}\left(X_{k-1}^{\left(j\right)}\right)dX_{k-1}^{\left(j\right)}}{r_{k|k-1}^{\left(j\right)}}\\ \;+\frac{p_{b,k}\left(X_{k}^{\left(j\right)}\right)\cdot \left(1-r_{k-1|k-1}^{\left(j\right)}\right)\cdot b_{k|k-1}\left(X_{k}^{\left(j\right)}\right)}{r_{k|k-1}^{\left(j\right)}} \end{array}$$
where $J_{k|k-1}=J_{P,k|k-1}+J_{B,k},J_{P,k|k-1}=J_{k-1}$. The number of Multi-Bernoulli parameter sets for survival sources and newborn sources are represented by $J_{P,k|k-1}$ and $J_{B,k}$, respectively. According to Equation (8), if the predicted Multi-Bernoulli parameter set can be expressed as ${\left\{{\hat{r}}_{k|k-1}^{\left(j\right)},{\hat{s}}_{k|k-1}\left(X_{k}^{\left(j\right)}\right)\right\}}_{j=1}^{J_{k|k-1}}$, then the update process can be expressed as
(15)$$p\left(X_{k}|Z_{1:k}\right)\approx {\left\{r_{k|k}^{\left(j\right)},s_{k|k}\left(X_{k|k-1}^{\left(j\right)}\right)\right\}}_{j=1}^{J_{k}}$$
where
(16)$$r_{k|k}^{\left(j\right)}=\frac{{\hat{r}}_{k|k-1}^{\left(j\right)}\int g\left(Z_{k}|X_{k}^{\left(j\right)}\right){\hat{s}}_{k|k-1}\left(X_{k}^{\left(j\right)}\right)dX_{k}^{\left(j\right)}}{1-{\hat{r}}_{k|k-1}^{\left(j\right)}+{\hat{r}}_{k|k-1}^{\left(j\right)}\int g\left(Z_{k}|X_{k}^{\left(j\right)}\right){\hat{s}}_{k|k-1}\left(X_{k}^{\left(j\right)}\right)dX_{k}^{\left(j\right)}}$$
(17)$$s_{k|k}\left(X_{k}^{\left(j\right)}\right)=\frac{g\left(Z_{k}|X_{k}^{\left(j\right)}\right){\hat{s}}_{k|k-1}\left(X_{k}^{\left(j\right)}\right)}{\int g\left(Z_{k}|X_{k}^{\left(j\right)}\right){\hat{s}}_{k|k-1}\left(X_{k}^{\left(j\right)}\right)dX_{k}^{\left(j\right)}}$$
where $g\left(Z_{k}|X_{k}\right)$ denotes the likelihood function. If the covariance of the general sensor array at time *k* in Gaussian noise environment is $R_{k}$, the likelihood function can be expressed as
(18)$$g\left(Z_{k}|X_{k}\right)=\frac{1}{\pi^{M}\det\left(R_{k}\right)}\exp\left(-{\left(Z_{k}-A\left(X_{k}\right)S_{k}\right)}^{H}R_{k}^{-1}\left(Z_{k}-A\left(X_{k}\right)S_{k}\right)\right)$$

The frame of Formula (18) is not held in impulse noise, so we propose to replace the likelihood function with a spatial spectrum method.

## 4. Improved Algorithm for Likelihood Function

In the practical engineering application, to guarantee the real-time and effectiveness of the estimation, the observation matrix of the array is obtained with a limited number of snapshots. Assuming L observations at time *k*, the array covariance matrix is calculated as ${\hat{R}}_{k}=X\left(t_{k}\right)X{\left(t_{k}\right)}^{H}/L$. We assume that the noise vector N(t) is independent to the target signal and has a S*α*S distribution with a characteristic exponent of *α*. From [25], if the array observation matrix $Z_{k}$ at time *k* is obtained, the FLOM matrix is defined as
(19)$$\psi_{i,j}=E\left\{Z_{i,j}\left(k\right){\left|Z_{j,i}\left(k\right)\right|}^{p-2}Z^{*}{}_{j,i}\left(k\right)\right\}\;1<p<\alpha \leq 2$$
where $\psi_{i,j}$ represents the (i,j)th element of $\Psi_{k}$, and ${\left(\cdot \right)}^{*}$ represents conjugate operation. The dimension of matrix $\Psi_{k}$ is M×M. In [25], the authors derived the form of the FLOM matrix as
(20)$$\Psi_{k}=a\left(\theta_{k}\right)R_{s}a^{H}\left(\theta_{k}\right)+rI_{M}$$
where $R_{s}$ and r represent the source and additive noise of the FLOM matrix, respectively. As can be seen from Equation (20), the (i,j)th FLOM matrix element is defined as
(21)$$\psi_{i,j}=\frac{\sum_{l=1}^{L}Z_{i}\left(k\right){\left|Z_{j}\left(k\right)\right|}^{p-2}Z_{j}{}^{*}\left(k\right)}{L}$$

Fractional moment *p* must satisfy 1<p<α≤2. The FLOM is used to replace the covariance matrix of the signal in impulse noise, and then the eigendecomposition is performed on $\Psi_{k}$ in the MUSIC algorithm to obtain the noise subspace $U_{n}$. The form of the FLOM-MUSIC spatial spectrum estimation function is
(22)$$g\left(Z_{k}|X_{k}\right)=P_{FLOM-MUSIC}^{}\left(X_{k}\right)={\left|\frac{1}{a^{H}\left(CX_{k}\right)U_{n}U_{n}{}^{H}a\left(CX_{k}\right)}\right|}^{\zeta }$$
where C=[1,0], and the $CX_{k}$ represents source azimuth information. a(⋅) is a space vector, and $\zeta \in R^{+}$ represents an exponential weighting of the spatial spectrum. The response of the traditional MUSIC spatial spectral beamformer in an impulse noise environment is distorted, which can result in a significant degradation in the performance of the resampling step. After being weighted, the particles can be moved to the high likelihood region to the resampling performance.

## 5. UT-MeMBer DOA Particle Filter Tracking Algorithm

In this section, we describe the particle filter implementation of the UT-MeMBer algorithm. From [22], if the multi-target probability density parameter set at time k−1 is ${\left\{\left(r_{k-1|k-1}^{j},s_{k-1|k-1}^{j}\right)\right\}}_{j=1}^{J_{k-1}}$, then the spatial posterior probability density at time k−1 and can be expressed as:(23)$$s_{k-1|k-1}^{\left(j\right)}\left(x\right)=\sum_{i=1}^{N_{k-1}}\omega_{k-1}^{\left(i,j\right)}x_{k-1}^{\left(i,j\right)},j=1,\ldots ,J_{k-1}$$
where $s_{k-1|k-1}^{j}$ is the spatial posterior probability density, which can be approximated as the weighted particle set ${\left\{\omega_{k-1}^{\left(i\right)},x_{k-1}^{\left(i\right)}\right\}}_{i=1}^{N_{k-1}}$. $N_{k-1}$ is the total number of particles, where $x_{k-1}^{\left(i\right)}$ represents the state of the *i* th particle, including angle and speed, i.e., $x_{k-1}^{\left(i\right)}={\left[\theta_{k-1},{\dot{\theta }}_{k-1}\right]}^{T}$. $\omega_{k-1}^{\left(i\right)}$ denotes the weight, usually satisfying $\sum_{i=1}^{N_{k-1}}\omega_{k-1}^{\left(i\right)}=1$. 

According to (12), the spatial posterior probability density of the prediction step consists of two items and can be written as
(24)$$s_{k|k-1}^{\left(j\right)}\left(x\right)=\sum_{i=1}^{N_{k|k-1}}\omega_{k|k-1}^{\left(i,j\right)}x_{k|k-1}^{\left(i,j\right)},j=1,\ldots ,J_{k|k-1}$$
where $N_{k|k-1}=N_{k-1}+N_{B,k}$ and $J_{k|k-1}=J_{p,k|k-1}+J_{B,k}$ represent the number of predicted particles and predicted MeMBer parameter sets, respectively. All particles can be sampled from two parts:(25)$$x_{k|k-1}^{\left(i,j\right)}=\left\{ \begin{array}{ll} x_{k-1,UT}^{\left(i,j\right)}, & i=1,\ldots ,N_{k-1}\\ \beta_{k}\left(x_{k}|Z_{k-1}\right), & i=N_{k-1}+1,\ldots ,N_{k-1}+N_{B,k} \end{array}\right.$$

Among them, $N_{B,k}$ denotes the number of newborn particles at time *k*, $x_{k-1,UT}^{\left(i,j\right)}$ is obtained by UT of $x_{k|k-1}^{\left(i,j\right)}$ [13]. Particle filtering suffers from missing sample diversity, resulting in depletion of the sampled particles. In order to solve this problem, the surviving particles will be subjected to UT operations. A set of sigma points with different weights are selected by UT operation, and then the posterior probability density of the state is approximated by these sigma points. The weight is
(26)$$\omega_{k|k-1}^{\left(i,j\right)}=\{\begin{array}{c} \frac{p_{s}r_{k-1|k-1}^{\left(j\right)}}{r_{k|k-1}^{\left(j\right)}}\cdot \frac{f_{k|k-1}\left(x_{k|k-1}^{\left(i,j\right)}|x_{k-1|k-1}^{\left(i,j\right)}\right)}{\rho_{k}\left(x_{k|k-1}^{\left(i,j\right)}|x_{k-1|k-1}^{\left(i,j\right)},Z_{k}\right)}\cdot \omega_{k-1}^{\left(i,j\right)},\;\;i=1,\ldots N_{k-1}\\ \frac{p_{b}\left(1-r_{k-1|k-1}^{\left(j\right)}\right)}{r_{k|k-1}^{\left(j\right)}}\cdot \frac{b_{k|k-1}\left(x_{k|k-1}^{\left(i,j\right)}\right)}{\beta_{k}\left(x^{\left(i,j\right)},Z_{k-1}\right)}\cdot \frac{1}{B},\;\;i=N_{k-1}+1,\ldots ,N_{k-1}+B \end{array}$$
where $p_{s}$ and $p_{b}$ represent the survival probability of particles and the newborn probability of particles, respectively. $N_{k-1}$ is the number of surviving particles sampled from the transition probability density $f_{k|k-1}$, and B is the number of newborn particles from the proposal probability density $\beta_{k}$. If the prediction MeMBer parameter sets can be expressed as ${\left\{r_{k|k-1}^{j},{\left\{\omega_{k|k-1}^{\left(i,j\right)},x_{k|k-1}^{\left(i,j\right)}\right\}}_{i=1}^{N_{k|k-1}}\right\}}_{j=1}^{J_{k|k-1}}$ at time *k*, then the update MeMBer parameter sets can be written as ${\left\{r_{k|k}^{j},{\left\{\omega_{k}^{\left(i,j\right)},x_{k}^{\left(i,j\right)}\right\}}_{i=1}^{N_{k}}\right\}}_{j=1}^{J_{k}}$. The weight is
(27)$$\omega_{k}^{\left(i,j\right)}=p_{D,k}\left(x_{k|k-1}^{\left(i,j\right)}\right)\cdot g\left(Z_{k}|x_{k|k-1}^{\left(i,j\right)}\right)\cdot \omega_{k|k-1}^{\left(i,j\right)}$$
where $p_{D,k}$ is the detection probability, and the likelihood function $g\left(Z_{k}|x_{k|k-1}^{\left(i,j\right)}\right)$ calculated by the MUSIC algorithm can be expressed as
(28)$$g\left(Z_{k}|x_{k|k-1}^{\left(i,j\right)}\right)=P_{FLOM-MUSIC}\left(Cx_{k|k-1}^{\left(i,j\right)}\right)={\left|\frac{1}{a{\left(Cx_{k|k-1}^{\left(i,j\right)}\right)}^{H}U_{n}U_{n}^{H}a\left(Cx_{k|k-1}^{\left(i,j\right)}\right)}\right|}^{\zeta }$$
where C=[1,0], and $Cx_{k|k-1}^{\left(i,j\right)}$ represents the azimuth angle information, ζ is the exponential weighting factor. $U_{n}$ represents the noise subspace obtained by the MUSIC algorithm. The steps of the UT-MeMBer DOA particle filter tracking algorithm are shown in Algorithm 1.

**Algorithm 1** UT-MeMBer DOA particle filter tracking algorithm
**Input:**
$\left[{\left\{r_{k-1|k-1}^{\left(j\right)},{\left\{\omega_{k-1}^{\left(i,j\right)},x_{k-1}^{\left(i,j\right)}\right\}}_{i=1}^{N_{k-1}}\right\}}_{j=1}^{J_{k-1}},Z_{k}\right]$

**Time Update**
1. Predict the existence probability: rk|k−1j=rP,k|k−1j+rB,kj.
where rP,k|k−1j=rk−1j⋅∑i=1Nk−1ωk−1(i,j)⋅ps,k(xk−1(i,j)) denotes the existence probability of survival model, rB,kj=(1−rk−1j)⋅∑i=1NB,kωk−1(i,j)⋅pb,k(xk−1(i,j)) represents the existence probability of newborn model. 2.Calculate the predicted state of surviving particles: [{xk|k−1(i,j)}i=1Nk−1]=UT[{xk−1(i,j)}i=1Nk−1].
  -Calculate the array flow matrix A(Cxk−1(i,j));
  -Calculate the amplitude of the signal S=[A(θ)HA(θ)]−1A(θ)HZk;
  -Calculate the noise variance σ2=1P∑p=1P|Zk−A(θ)S|2;
  -Select a weighted sample point of 2nx+1
for each particle xk−1(i,j), where         χ0=xk−1(i,j),W0=κ/(nx+κ)s=0χs=xk−1(i,j)+((nx+κ)σ2),Ws=κ/2(nx+κ)s=1,…,nxχs=xk−1(i,j)−((nx+κ)σ2),Ws=κ/2(nx+κ)s=nx+1,…,2nx,        
κ=5 is a secondary scaling parameter, nx=2.
  -Each sigma point propagates through a nonlinear function: γs=fk|k−1(χs),s=1,…,2nx;
  -Compute the mean and covariance of γs: ψ¯=∑s=02nxWsγs,P=∑s=02nxWs(γs−ψ¯)(γs−ψ¯)T;
  -Obtain: xk|k−1(i,j)∼N(ψ¯,P);3.Construct a newborn target weighted particle: xk|k−1(i,j)∼βk(xk|Zk−1),i=Nk−1+1,…,Nk−1+NB,k.4.Calculate the prediction weight ωk|k−1(i,j),i=1,…,Nk|k−1 according to (26).5.Unite weighted particle set: {(xk|k−1(i,j),ωk|k−1(i,j))i=1Nk|k−1}j=1Jk|k−1={(xp,k−1(i,j),ωp,k−1(i,j))i=1Nk−1}j=1Jk−1∪{(xB,k(i,j),ωB,k(i,j))i=1NB,k}j=1JB,kwhere Jk|k−1=Jk−1+JB,k, Nk|k−1=Nk−1+NB,k.
**Measurements Update**
6.For each particle xk|k−1(i,j), Calculate the likelihood function g(Zk|xk|k−1(i,j)) according to (28).7.Update existence probability:rk|kj=rk|k−1j⋅∑i=1Nk|k−1g(Zk|xk|k−1(i,j))ωk|k−1(i,j)pD,k(xk|k−1(i,j))1−rk|k−1j+rk|k−1j⋅∑i=1Nk|k−1g(Zk|xk|k−1(i,j))ωk|k−1(i,j)pD,k(xk|k−1(i,j)). where j=1,⋯,Jk|k−1.8.The updated weight is calculated by (27) and normalized ωk(i,j)=ω˜k(i,j)/(∑j=1Jk|k−1∑i=1Nk|k−1ω˜k(i,j)).
**Resample Step**
9.${\left\{{\left(x_{k|k-1}^{\left(i,j\right)},\omega_{k|k-1}^{\left(i,j\right)}\right)}_{i=1}^{N_{k|k-1}}\right\}}_{j=1}^{J_{k|k-1}}\to {\left\{{\left(x_{k}^{\left(i,j\right)},\omega_{k}^{\left(i,j\right)}\right)}_{i=1}^{N_{k}}\right\}}_{j=1}^{J_{k}}$.**Output**: ${\left\{r_{k}^{j},{\left(x_{k}^{\left(i,j\right)},\omega_{k}^{\left(i,j\right)}\right)}_{i=1}^{N_{k}}\right\}}_{j=1}^{J_{k}}$.

Algorithm 1 gives the pseudo-code of UT-MeMBer DOA particle filter tracking algorithm. The prediction is made in steps 1–5. Step 6 calculates each predicted particle likelihood function which is replaced by the MUSIC spatial spectral function. The update existence probability is calculated in step 7. Step 8 calculates the normalized weight. Particle resampling is performed in step 9. The particle set ${\left\{{\left\{\omega_{k}^{\left(i,j\right)},x_{k}^{\left(i,j\right)}\right\}}_{i=1}^{N_{k}}\right\}}_{j=1}^{J_{k}}$ approximates the spatial probability density function $s_{k|k}^{j}$, and the estimation of updated source can be expressed as ${\bar{x}}_{k}=\sum_{i=1}^{N}\omega_{k}^{\left(i,j\right)}\cdot x_{k}^{\left(i,j\right)}$. 

## 6. Simulation Results

Since the traditional MUSIC algorithm cannot solve the multi-source tracking problem when target number is varying, this paper uses FLOM matrix to substitute the covariance matrix to obtain the corresponding MUSIC spatial spectrum, which can be as the particle likelihood function. We proposed a UT-MeMBer DOA tracking algorithm under RFS framework, which can be named as UT-MB-FLOM-MUSIC algorithm. The Generalized Signal to Noise Ratio (GSNR) is defined as
(29)$$\mathrm{GSNR}=10\log\left(E\left\{\;{\left|s\left(k\right)\right|}^{2}\right\}/\gamma \right)$$
where γ represents the noise dispersion parameter, and GSNR represents the ratio of signal intensity and noise dispersion. In the simulation, different characteristic indices *α* describe the degree of impact of different noises.

In the following simulation experiments, the estimated performance is evaluated by the root mean square error (RMSE), which is defined as
(30)$$\mathrm{RMSE}=\frac{1}{P}\sum_{p=1}^{P}\frac{1}{MC}\sum_{j=1}^{MC}\left(\sqrt{\frac{1}{K}\sum_{i=1}^{K}{\left(x_{ij}-{\bar{x}}_{ij}\right)}^{2}}\right)$$
where $x_{ij}$ and ${\bar{x}}_{ij}$ represent the estimated values and real values of the azimuth angle in the *j*th Monte Carlo (MC) simulation experiment at time *i*, respectively, and *P* indicates the number of sources at time *i*.

Assuming that the sources $x_{k}={\left[\theta_{k}\left(t\right),{\dot{\theta }}_{k}\left(t\right)\right]}^{T}$ move with a constant velocity ${\dot{\theta }}_{k}\left(t\right)$ rad/s, the constant velocity (CV) model is given as
(31)$$x_{k}=F_{k}x_{k-1}+Gv_{k}$$
where the transfer matrix $F_{k}$ and G are defined by
(32)$$F_{k}=\left[\begin{array}{cc} 1 & \Delta T\\ 0 & 1 \end{array}\right];\;G=\left[\begin{array}{c} \Delta T^{2}/2\\ \Delta T \end{array}\right]$$
respectively, where ΔT=1s denotes the time step, and $v_{k}$ is a zero-mean real Gaussian process used to model the disturbance on the source velocity, i.e., $v_{k}\sim N\left(0,\Sigma_{k}\right)$ with $\Sigma_{k}=1$.

Experimental conditions are as follows: The number of array elements is M=10, d=λ/2, the observation time is K=50s, *L* = 100, GSNR = 10 dB, MC = 100, and ξ=5. The source survival probability $p_{s,k}\left(x_{k}\right)=0.99$, and the source detection probability $p_{D,k}\left(x_{k}\right)=0.98$. In the UT-MB-FLOM-MUSIC algorithm prediction step, we assume that there are six new sources at each time, i.e., $J_{B,k}=6$, all obeying a uniform distribution on [−π/2 , π/2] and each new source produces 300 particles, i.e., $N_{B,k}=300$. In the update step, the MUSIC spatial spectral function is used to replace the likelihood function and is exponentially weighted, which improves the feasibility of the algorithm. In the impulse noise model, the noise is Gaussian noise when *α* = 2. The DOA estimation method based on the MeMBer can be named as MB-MUSIC algorithm, and the DOA estimation method based on the MeMBer of FLOM vector can be named as MB-FLOM-MUSIC algorithm.

### 6.1. Scenario 1: The Number of Targets Is Not Time-Varying

Consider a linear multi-source scenario with two sources. Since the PASTD algorithm cannot track the time-varying target, all the target survival time are 1–50 s. The initial source state are $x_{1}=\left[-30;-0.5\right]$, and $x_{2}=\left[5;0.5\right]$.

Figure 1a shows the RMSE of angles for four algorithms when running 100 MC at *α* = 2, GSNR = 10 dB, and Figure 1b shows two source trajectories for a single MC. It can be seen from Figure 1a that the UT-MB-FLOM-MUSIC algorithm proposed in this paper is obviously better than the traditional PASTD and has the highest accuracy when the number of targets is constant. It can be seen in Figure 1b that the algorithm can effectively track the target trajectory, while the PASTD algorithm deviates from the real trajectory at several times.

We show the RMSE for tracking the multi-source motion when *α* = 1.3, GSNR = 10 dB, MC = 100, and *L* = 100 in Figure 2a. It can be seen from Figure 2a that the RMSE of the UT-MB-FLOM-MUSIC algorithm is smaller than that of the other three algorithms. The accuracy of the MB-MUSIC algorithm is significantly reduced in impulse noise, and the PAST algorithm is more accurate than MB-MUSIC. It can be seen from Figure 2b that the MB-MUSIC algorithm cannot effectively track the target trajectory in impulse noise, and the PASTD algorithm also has the problem of inaccurate target tracking. Based on the fact that the above target numbers are unchanged, we will analyze the target time-varying DOA tracking.

### 6.2. Scenario 2: The Number of Targets Is Time-Varying

Consider a linear multi-source scenario with three sources. The number of sources is time-varying due to births and deaths, the survival time of the four sources is 1–50 s, 10–50 s, 20–45 s, and the initial source states are $x_{1}=\left[-30;-0.5\right]$, $x_{2}=\left[5;1.0\right]$, and $x_{3}=\left[60;-2.0\right]$. 

Figure 3a shows the RMSE of angles for three algorithms for running 100 MC at *α* = 2, *L* = 100 and GSNR = 10 dB, and Figure 3b shows three sources trajectory for a single MC. It can be seen from Figure 3 that the likelihood function of the MUSIC spatial spectrum instead of the Multi-Bernoulli particle filter update stage can effectively estimate the target number and motion state, and also verify the feasibility of the literature [14] in the Gaussian noise environment. Although the error is large at time 35, the overall error is below 2 degrees. It can also be seen from Figure 3a that the RMSE of the UT-MB-FLOM-MUSIC algorithm is also smaller than other algorithms even in the Gaussian noise environment.

Since Gaussian noise does not reflect true signal interference, the *α* stable distribution can reflect the impact of impulse noise. Figure 4 shows the RMSE and cardinality estimation error plots for three algorithms running 100 MC when the characteristic index *α* is different and the GSNR = 10 dB, *L* = 100. It can be seen from Figure 4a that, in α=1.1~1.9, the RMSE error of the three estimation algorithms first decreases, and finally tends to be flat. It also can be seen that the RMSE of the UT-MB-FLOM-MUSIC algorithm is significantly smaller than the MB-FLOM-MUSIC and MB-MUSIC algorithms when *α* = 1.1 or *α* = 1.2, so that the UT-MB-FLOM-MUSIC algorithm has a better effect when handling the impulse noise environment. Since the characteristic index is close to 2 when *α* = 1.8 or *α* = 1.9, Figure 4b shows that the cardinality estimation error of the three algorithms approaches 0. It also shows that it is feasible to use the MUSIC spatial spectrum as a substitute for the likelihood function when the noise environment is close to Gaussian noise while the MUSIC algorithm cannot effectively estimate the number of targets in an impulse noise environment.

In Figure 5, we show the RMSE and cardinality estimation for tracking the multi-source motion when *α* = 1.3 and GSNR = 10 dB, MC = 100. It can be seen from Figure 5 that the RMSE of the UT-MB-FLOM-MUSIC algorithm is smaller than that of the other two algorithms. Although the RMSE will increase when the new target appears or disappears, it will decrease rapidly at the next time step. This phenomenon shows that the Multi-Bernoulli filter has a large recognition performance for the target and can quickly track the state of the target. Table 1 shows the RMSE and computing performance of the MB-MUSIC algorithm, MB-FLOM-MUSIC algorithm and the UT-MB-FLOM-MUSIC algorithm at one MC.

The operating environment includes an Intel (R) Core (TM) i5-8500 CPU @ 3.00 GHz processor and a 64-bit operating system MATLAB 2014. It can be seen from Table 1 that the UT-MB-FLOM-MUSIC algorithm RMSE is smaller than other algorithms when the running time is too long.

Figure 6 analyzes the RMSE and probability of convergence (PROC) for three algorithms running 100 MC when *α* = 1.3 and GSNR = 0–16 dB. where $\mathrm{PROC}=\frac{1}{K}\sum_{i=1}^{K}\sum_{j=1}^{MC}1_{ij}/MC\times 100\%$, and $1_{ij}$ is defined as $1_{ij}=\left\{ \begin{array}{l} 1,\left|x_{ij}-{\bar{x}}_{ij}\right|<\varepsilon \\ 0,otherwise \end{array}\right.$. let ε=1. It can be seen from Figure 6a that the MB-FLOM-MUSIC and UT-MB-FLOM-MUSIC algorithms have higher accuracy than the MB-MUSIC in an impulse noise environment, and the UT-MB-FLOM-MUSIC algorithm has higher accuracy under the high GSNR. It can be seen form Figure 6b that as the SNR increases, the PROC increases. And at the same GSNR, the performance of the MB-FLOM-MUSIC algorithm is more significant.

Figure 7 shows the RMSE of three algorithms running 100 MC when *α* = 1.3 and the snapshot number *L* = 50, 100, 150. It can be seen that the UT-MB-FLOM-MUSIC algorithm has the smallest RMSE and it works best when the snapshot number is *L* = 150.

### 6.3. Scenario 3: The Number of Targets Is Time-Varying and Maneuvering

Consider a nonlinear multi-source scenario with three sources. The number of sources is time-varying due to births and deaths, and the survival time of the three sources is 1–50 s, 10–50 s, 20–45 s, and the initial source state are $x_{1}=\left[-30;-0.5\right],$
$x_{2}=\left[5;1.8\right]$, and $x_{3}=\left[60;-2.0\right]$. The state transition matrix of the collaborative turning (CT) model is
(33)$$F_{k}=\left[\begin{array}{cc} 1 & \sin\left(T\omega \right)/\omega \\ 0 & \cos\left(T\omega \right) \end{array}\right]$$
where ω=0.25 rad and other experimental conditions are the same as scenario 1.

Figure 8 shows the maneuvering target trajectory of three algorithms running one MC when *α* = 1.3, *L* = 100, and GSNR = 10 dB. It can be clearly seen from Figure 8 that the three methods lose the target when the target crosses at time 33, but after time 36, the MB-FLOM-MUSIC algorithm and the UT-MB-FLOM-MUSIC algorithm can still capture the target state. Compared with the MB-FLOM-MUSIC algorithm, the target state estimation value of the UT-MB-FLOM-MUSIC algorithm is closer to the true value.

In Figure 9, we show the RMSE and cardinality estimation for tracking the multi-source motion when *α* = 1.3 and GSNR = 10 dB, MC = 100. It can be seen from Figure 9a that the RMSE of the UT-MB-FLOM-MUSIC algorithm is smaller than that of the other two algorithms. As can be seen from Figure 9b, when the target is maneuvering, the target is not captured by the three algorithms from time 33, but after time 36, the MB-FLOM-MUSIC algorithm and UT-MB-FLOM-MUSIC algorithm can still estimate the number of targets in time. Compared with the result of Figure 5b, the performance to capture targets of the UT-MB-FLOM-MUSIC algorithm is significantly weakened.

Table 2 shows the RMSE and computing performance of the MB-MUSIC algorithm, MB-FLOM-MUSIC algorithm and the UT-MB-FLOM-MUSIC algorithm. Compared with the results in Table 1, the RMSE and running time of the three algorithms are increased when the target is maneuvered. The RMSE of UT-MB-FLOM-MUSIC algorithm is smaller than other two algorithms when the running time is long.

## 7. Conclusions

A DOA tracking algorithm based on the UT-MeMBer particle filter in an impulse noise environment is proposed in this paper. Since the FLOM matrix is used instead of the covariance matrix, the spatial spectrum based on FLOM can well reflect the real DOA in impulse noise environment. For the persistent surviving particles, the sigma point is selected by UT to approximate the posterior density of the state to improve the performance of the particle. Then, the MUSIC spatial spectral function of the FLOM matrix is used to represent the likelihood function of the particle. And the weighting of the likelihood function can further increase the weight of the particles in the high likelihood region. The results show that the UT-MB-FLOM-MUSIC algorithm is more effective than the PASTD, MB-MUSIC, and MB-FLOM-MUSIC algorithms in an impulse noise environment. The advantage of this algorithm is that the MeMBer filter can operate the array data more directly, and can effectively track the target number of time-varying DOA. The shortcoming of this algorithm is that it takes a long time. Our future work will focus on how to improve the efficiency of the algorithm, maneuvering target tracking in other noisy environments, etc. 

## Figures and Tables

**Figure 1 sensors-19-04031-f001:**
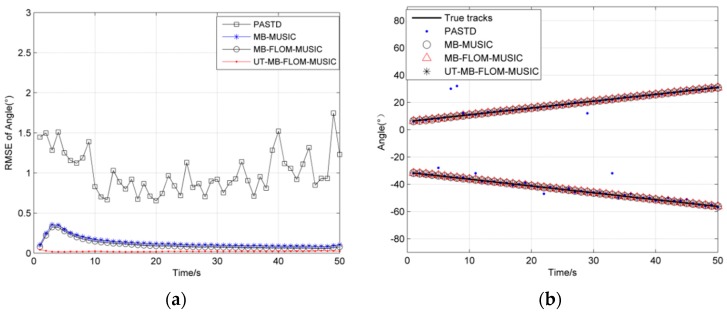
Root mean square error (RMSE) of angle under *α* = 2, *L* = 100 and Generalized Signal to Noise Ratio (GSNR) = 10 dB: (**a**) The RMSE of 100 MC; (**b**) source trajectory of Single MC.

**Figure 2 sensors-19-04031-f002:**
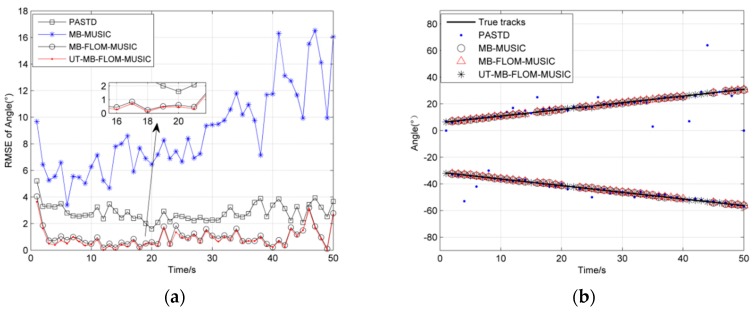
RMSE of angle under *α* = 1.3, *L* = 100 and GSNR = 10 dB: (**a**) The RMSE of 100 MC; (**b**) source trajectory of Single MC.

**Figure 3 sensors-19-04031-f003:**
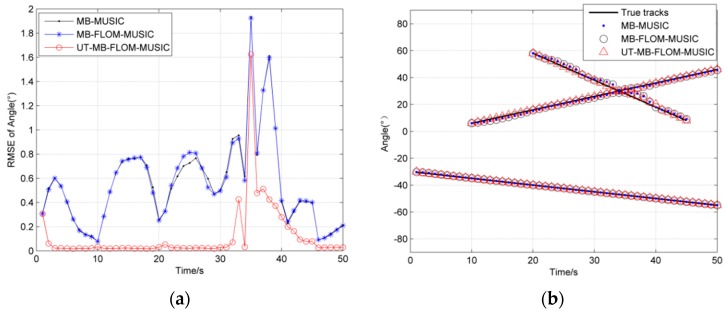
RMSE of angle under *α* = 2, *L* = 100 and GSNR = 10 dB: (**a**) The RMSE of 100 MC; (**b**) source trajectory of Single MC.

**Figure 4 sensors-19-04031-f004:**
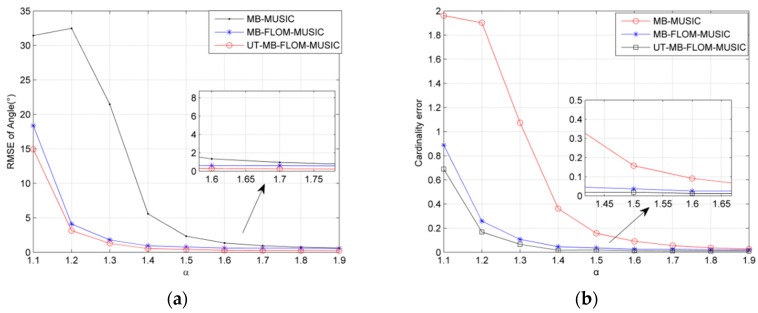
RMSE and cardinality error of angle under different *α*, *L* = 100, MC = 100 and GSNR = 10 dB: (**a**) The RMSE under different *α*; (**b**) cardinality error under different *α*.

**Figure 5 sensors-19-04031-f005:**
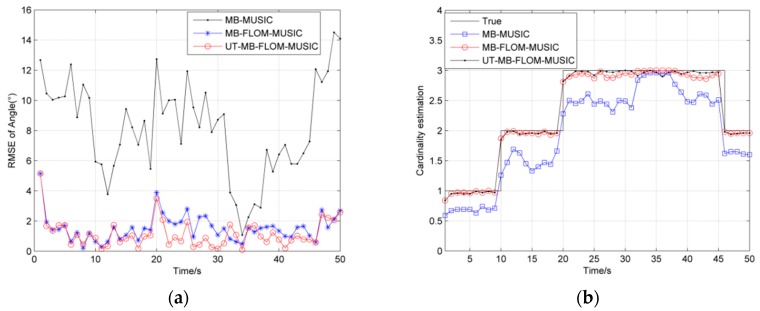
RMSE and Cardinality estimation of angle under *α* = 1.3 and GSNR = 10 dB, MC = 100: (**a**) RMSE of angle; (**b**) Cardinality estimation of angle.

**Figure 6 sensors-19-04031-f006:**
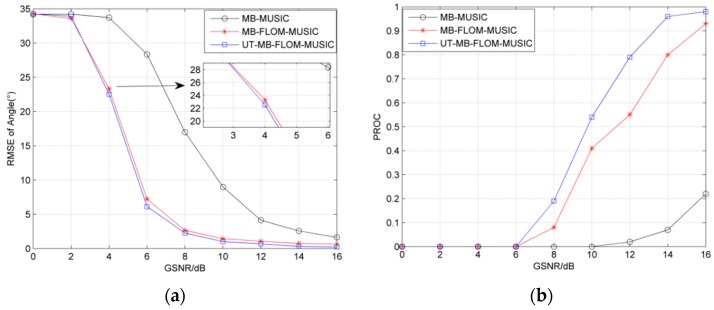
RMSE and probability of convergence (PROC) of the angle under different GSNR, *α* = 1.3 MC = 100 and *L* = 100: (**a**) RMSE of angle; (**b**) PROC at different GSNR.

**Figure 7 sensors-19-04031-f007:**
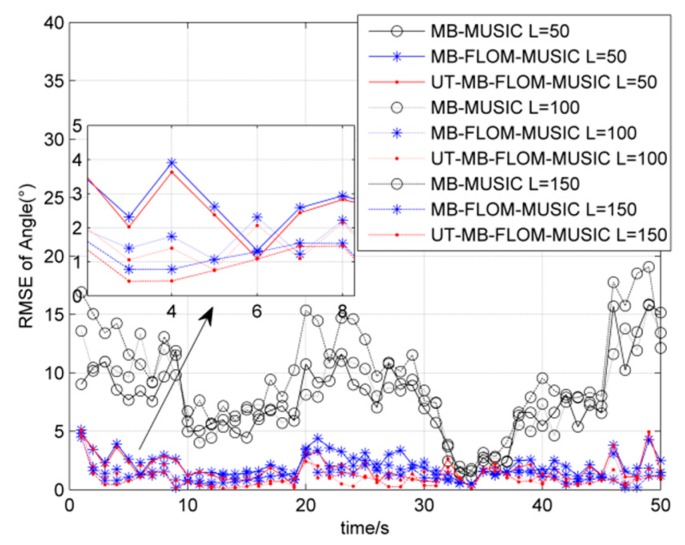
RMSE for source tracking under different snapshots, *α* = 1.3 MC = 100 and GSNR = 10 dB.

**Figure 8 sensors-19-04031-f008:**
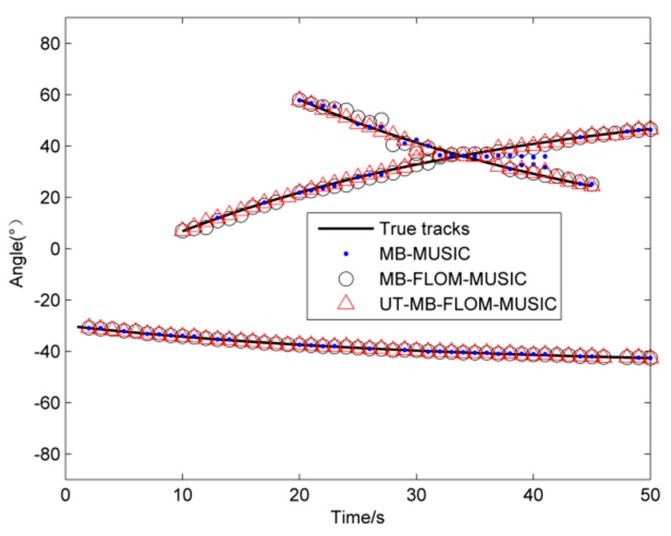
Target trajectory, *α* = 1.3, L = 100, and GSNR = 10 dB.

**Figure 9 sensors-19-04031-f009:**
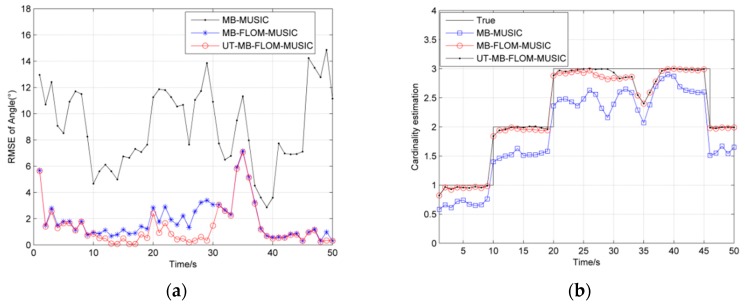
RMSE and cardinality estimation of angle under *α* = 1.3 and GSNR = 10 dB, MC = 100: (**a**) RMSE of angle; (**b**) cardinality estimation of angle.

**Table 1 sensors-19-04031-t001:** Running Time (CV model).

Algorithm	RMSE	Running Time/s
MB-MUSIC	7.6012	2.94
MB-FLOM-MUSIC	1.1396	9.59
UT-MB-FLOM-MUSIC	0.2698	114.67

**Table 2 sensors-19-04031-t002:** Running Time (CT model).

Algorithm	RMSE	Running Time/s
MB-MUSIC	8.7728	3.67
MB-FLOM-MUSIC	1.3198	10.73
UT-MB-FLOM-MUSIC	0.6102	135.30

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
