# Peer review of "DOA Tracking Based on Unscented Transform Multi-Bernoulli Filter in Impulse Noise Environment"

_sensors, 2019, doi:10.3390/s19184031_

Round 1

Reviewer 1 Report

See attachment.

Reviewer 2 Report

Authors proposed a DOA tracking algorithm based on UT-MeMBer particle filter in impulse noise environment in this manuscript.  The advantage of the implemented algorithm lies in the sense that the MeMBer filter can operate the array data more directly, and can effectively track the target number of time-varying DOA. The shortcoming of this algorithm as proposed by authors is that it takes a long time. 

Since authors claim that the future work will focus on more complex situations, however, I strongly believe that authors need to include it in the current version in order to see the algorithm superiority and limitation in more defined scenarios.

Scenario 1 and 2 are not enough to claim contribution. Needs to be explored further as recommended above.

Quality of all figures need to be improved and traces need be thicker than current. Also, scaling should be defined in a smarter way in order that the main information is more clearer.

The manuscript can be enriched by introducing the recent references in the introduction section.

A detailed comparison table need to be provided at the end in order to show the performance comparison with other algorithms including MUSIC, MB MUSIC, etc

Majority of basic equations are not referenced and need to be referenced. the one that manuscript has self-contained is fine.

Line 106 and 107 have formatting issue. please correct it.

for algorithm 1 it will be interesting if it is explained in terms of the block diagram flow chart.

Round 2

Reviewer 1 Report

The reviewer appreciates the author's revision to my satisfactory. No further comments.

Reviewer 2 Report

thank you for revising the manuscript.

All my comments are addressed well except the one regarding flowchart.

What I was thinking that it will create ease for readers, anyhow I am also fine with the current style.

Thank you for adding and analyzing additional scenarios as well.

I recommend the manuscript in the current form.